# Development of the Italian version of the Orgasmic Perception Questionnaire (OPQ)

**Marta Panzeri**[1]*, **Denise Mauro**[2], **Lucia Ronconi**[3], **Ana Isabel Arcos-Romero**[4]

**1** Department of Developmental Psychology and Socialisation, Padua University, Padova, Italy, **2** Department of General Psychology, Padua University, Padova, Italy, **3** FISPPA–Università degli studi di Padova, Padova, Italy, **4** Department of Psychology, Loyola University, Sevilla, Spain

* marta.panzeri@unipd.it

**Data Availability Statement:** All data used in this study can be found in the Research Data UNIPD at https://researchdata.cab.unipd.it/id/eprint/910 [DOI: 10.25430/researchdata.cab.unipd.it. 00000910].

## Abstract

Orgasm is a phase of the human sexual response, and the possible discrepancies between male and female ways to experience it are still not clear in the literature. There is a lack of tools to adequately assess orgasm perception. This study aims to develop an instrument and verify possible differences between males and females. We constructed the Orgasmic Perception Questionnaire (OPQ) through different stages: first, 316 items selection was conducted on a sample of 96 people, where items came mainly from written descriptions of orgasm perception; second, an exploratory factor analysis was conducted on 674 Italian adults with a 63-item OPQ; finally, a confirmatory factor analysis was conducted on 1100 Italian adults with a 47-item OPQ. In the first study, 63 items fitted an equidistributional pattern and were to form the 63-item OPQ used for EFA. The EFA showed that five factors out of 47 explained 44.01% of the total variance and were named: Ecstasy, Contractions, Relaxation, Power, and Sensations. The confirmatory factor analyses run on the 47-item OPQ confirmed that the five-factor structure fits. Moreover, females scored higher than males with an adequate effect size in two factors: Contractions and Sensations. In conclusion, the OPQ could be a useful tool in both clinical settings and research studies to investigate the perception of orgasmic experience in its totality.

## Introduction

There is no one agreed-upon definition of orgasm. In the literature, numerous studies have attempted to formulate with little success a universally accepted definition of orgasm comprehensive and inclusive of the many aspects that characterize this experience and the variables related to it [1]. Orgasm appears to be the most complex and least understood event among sexual responses [2, 3]. The orgasm experience is the perception of the psychological sensations of orgasm [4]. Orgasm constitutes an intense sensation of pleasure that is accompanied by an alteration in consciousness, intense and quick contractions of the genitourinary musculature [5], and an increase in the respiratory rate, blood pressure, and heart rate [2]. It is a complex mixture of subjective mental and physical body changes [6].

Considering human orgasm as a biopsychosocial experience, Mah and Binik [1] developed and evaluated a new multidimensional descriptive model and methodology that integrates a

**Funding:** The authors received no specific funding for this work.

**Competing interests:** The authors have declared that no competing interests exist.

biological and psychological perspective. This approach, applicable to both genders, aims to systematically investigate both the essential characteristics and the variability of the orgasm experience. In Mah and Binik's orgasm model, the sensory, affective, and evaluative dimensions are considered in a multidimensional framework. Each dimension includes different components of the orgasmic experience: the sensory dimension includes the perception of physiological events (such as contractile sensations, muscle tension and release, and thermal sensations); the affective dimension concerns the positive and negative emotions felt during orgasm (such as euphoria, intimacy, love); and the evaluative dimension refers to evaluations about the orgasmic experience (intensity, pleasure, satisfaction, pain). The model suggests that all orgasm experiences involve all three dimensions and may vary along these. The descriptions of orgasm found in the literature concern multiple aspects inherent in the psychological experience, the physiological changes related to the body, and the genital sensations: varying both from person to person and within the experiences experienced by a single individual [7].

Literature is still unambiguous about gender differences: some authors emphasize the similarity of male and female orgasms [7–9], while others state a disparity from both a physiological and psychological point of view, considering them two separate phenomena. Gender differences have been demonstrated in the perception of psychological [10, 11] and physiological sensations of orgasm [12]. Despite less attention paid to male orgasm, literature found that orgasm experiences in men are influenced by psychosocial aspects [13, 14] as well as in women, for whom orgasmic satisfaction and pleasure are related also to intrapersonal, interpersonal, and contextual factors [15–17]. From a physiological point of view, Masters and Johnson [8] found minimal differences between men and women concerning the orgasmic experience: in men, the orgasmic experience proceeds even if the stimulus that produced it ceases, whereas in women if the excitatory stimulus disappears the orgasm itself also ends. It takes little to stop the female orgasm, while the male orgasm goes on [8, 18]. In any case, Masters and Johnson [8] consider both phenomena to be biologically indistinguishable. Orgasm duration seems to be longer in males (average contractile duration 25 seconds [19]) than in females (16 seconds [20]). The literature on brain imaging during orgasm in men and women is relatively limited and variable [21].

In 1976, Vance and Wagner [7] conducted a qualitative study to verify a potential difference between the male and female orgasmic experience: 40 judges (obstetrician-gynecologists, psychologists, and medical students) were to sex-identify 96 descriptions of orgasm written by college students. Judges were unable to identify the gender of the authors of the written description, contrary to Vance and Wagner's hypothesis.

To evaluate the orgasm experience, some instruments have been developed. Mah and Binik [11] developed the Orgasm Rating Scale (ORS) to specifically assess the psychological characteristics of the orgasm using adjectives to describe its qualities in two contexts: solitary masturbation and sexual relationship. The Spanish version of this scale [22] examines the subjective orgasm experience along with 25 adjectives words that are rated according to the best description of the last orgasm experienced. The ORS encompasses two dimensions: the Sensory Dimension and the Cognitive/Affective Dimension. In their first study, Mah & Binik [11] found that the two-dimensional model was superior both to a one-dimensional that to a three-dimensional model, while the Spanish version [22] encompassed four dimensions (affective, sensory, intimacy, and rewards). The phenomenological adjectives used in the ORS do not capture completely the specific bodily sensations nor the specific feelings that are associated with climax. Dubray et al. [12] developed the Bodily Sensations of Orgasm questionnaire, a self-report questionnaire to assess the bodily and physiological sensations of orgasm meant to cover topics not enclosed by the ORS. It is limited to physical cues and it lacks a confirmatory factorial analysis to test the factorial model. Mollaioli et al. [23] created the "Orgasmometer", a

visual analog scale to evaluate the perception of orgasmic intensity in women. Although it is meant to be a very quick instrument, it is reductive in itself and can be useful as an add-on to a wider questionnaire. However, as far as we are aware, there are no previous instruments to examine complete descriptions of the bodily sensations, emotions, feelings, and perceptions experienced during orgasm for both men and women.

Thus, due to the lack of a standardized measure to specifically assess the complete experience of orgasm, the main objective of this study was to develop and validate the Orgasmic Perception Questionnaire (OPQ), based on written descriptions of lay people, since we thought that lay people can give us the most reliable information about the real orgasmic experience. Although the currently available instruments were first considered to carry out the adaptation and validation, as explained before, most of the scales focus on the evaluation of the sexual experience only in women, and none of them use all-inclusive descriptions of the orgasm. We decided to develop and validate a new questionnaire, based on lay people's real descriptions of their own orgasm experience, intending to embody the more as possible perceptions, emotions, feelings, and sensations of the orgasm with both a general and a clinical perspective. To do so, we run three stages study: Stage 1 aims at the design and construction of the questionnaire. In stage 2, we analyzed the exploratory Model. In stage 3, confirmatory factor analysis was carried out. In addition, the reliability of the instrument, the correlations between the factors, the convergent and divergent validity evidence, and the difference between males' and females' perceptions of the orgasm were tested.

## Stage 1: Construction of the questionnaire

### Method

**Item identification.**    The items were extrapolated from the written definitions of orgasm provided by the 364 undergraduate students (29.67% male and 70.33% female, over 18 years old). The participants were asked to write a description as detailed as possible of what an orgasm feels like, taking into account physical and psychological sensations, thoughts, and emotions. Each definition was analyzed by two researchers separately. From this first phase, 292 short sentences were obtained which made up the items of the questionnaire: sentences were analyzed one by one and modified in grammatical structure: they were placed in the first person; sentences that were alike or had the same meaning were eliminated. Sentences with a common meaning but different nuances were kept. Once the questionnaire was completed, 4 researchers in the sexology research were asked to revise the whole. Three out of four made some criticisms regarding the small number of items related to emotions. For this reason, brainstorming was done among four researchers, 2 males and 2 females: from the discussion, it emerged that the category of negative aspects related to orgasm was missing; several items on the topic have been proposed and 24 items were chosen to be added to the previous ones. The final version of the questionnaire encompassed 316 items (see the S1 File, OPQ-316) to be rated on a 5-point Likert scale ranging from "always" to "never" (see the S1 File, OPQ-316) with the addition of "I don't know/I don't remember". We administered two different versions to males and females due to the gender difference in the Italian language: the two versions differ only for the declination of nouns, adjectives, pronouns, articles, and past participles, such in "rilassato/rilassata" (relaxed) or "il/la" (the) for masculine/feminine. Items referred to different aspects of the orgasmic experience as sensorial ("I feel the warmth on my face"), emotional ("All anxiety disappears"), affective ("I experience a feeling of tranquility"), and relational ("I feel protected by the partner"). For this late category of items, we added the possibility to answer "I have never had a sexual partner".

**Table 1. Socio-demographical information.**

| | | Item identification Sample | EFA Analysis Sample | CFA Analysis Sample | Convergent Validity Sample | Divergent Validity Sample | Reliability Sample |
|---|---|---|---|---|---|---|---|
| Age range | Female | 18–60 | 18–60 | 18–59 | 18–51 | 19–59 | 19–51 |
| | Male | 18–64 | 18–64 | 18–63 | 18–51 | 19–59 | 19–51 |
| Mean | Female | 25.6 | 25.6 | 25.8 | 25.3 | 26.5 | 23.2 |
| | Male | 29.3 | 29.3 | 26.4 | 23.9 | 25.6 | 23.2 |
| SD | Female | 6.7 | 6.7 | 7.5 | 5.5 | 7.1 | 4.5 |
| | Male | 9.6 | 9.6 | 7.9 | 5.2 | 5.9 | 4.5 |
| Nationality | Italian | 99.0% | 100.0% | 97.9% | 97.7% | 85.1% | 87.5% |
| | Other | 1.0% | 0.0% | 2.1% | 2.3% | 14.9% | 1.4% |
| Civil status | Single—Unmarried | 78.1% | 76.6% | 67.3% | 91.3% | 43.4% | 91.3% |
| | Cohabiting—Married | 16.7% | 20.0% | 16.2% | 8.1% | 13.6% | 8.1% |
| | Separeted—Divorced | 5.2% | 2.4% | 1.7% | 0.6% | 1.5% | 0.5% |
| Education | Junior high school | 4.2% | 5.2% | 0.5% | 0.6% | 1.0% | 0.0% |
| | Professional degree | 3.1% | 3.9% | 2.3% | 1.7% | 2.0% | 1.6% |
| | High school | 41.7.% | 39.6% | 49.4% | 49.7% | 31.8% | 61.1% |
| | Bachelor/master | 51.0% | 44.5% | 39.7% | 40.2% | 57.1% | 33.5% |
| | Higher degree | 0.0% | 6.7% | 6.5% | 7.8.% | 8.1% | 3.2% |
| Sexual orientation | Completely heterosexual | 84.4% | 71.5% | 60.9% | 58.6% | 62.6% | 54.0% |
| | Predominantly heterosexual | 12.5% | 22.7% | 27.2% | 30.7% | 21.8% | 33.8% |
| | Bisexual | 2.1% | 2.2% | 4.5% | 5.6% | 7.1% | 6.9% |
| | Predominantly homosexual | 1.0% | 1.9% | 2.0% | 1.7% | 3.0% | 3.7% |
| | Completely homosexual | 0.0% | 1.6% | 3.5% | 3.4% | 4.0% | 1.6% |
| | Asexual | 0.0% | 0.0% | 0.4% | 0.0% | 1.5% | 0.0% |

**Participants.** Of the 100 participants sample, four were eliminated for not meeting the inclusion criteria. The final sample was composed of 96 Italian adults (45.8% men, 54.2% women), which socio-demographic characteristics as shown in Table 1.

**Procedure.** Participants had to rate the 316 selected items and sign the informed consent. A background questionnaire about age, gender, nationality, civil status, education level, employment, sexual orientation, relationship, and last sexual experience was administrated. We choose to not use quantitative content validity measures since we wished that the selection would be performed by data analysis and not by experts: there are still too many biases on orgasm that according to us would not allow experts to judge objectively. Moreover, we wished to select items that could represent different ways to experience orgasm more than items that represent how most people experience it, and that could emerge only from laypeople's judgment.

The collection of personal data provided for the possibility of not replying, by marking the option "I prefer not to reply", so socio-demographical data were not available for the whole sample (see Table 1). The questionnaire was administered on paper to the general Italian population. The inclusion criteria included being older than 18 years and Italian mother tongue. Participants were recruited in public places. Data were collected from May 2018 to January 2019 anonymously, so that authors had no access to information that could identify individual participants during or after data collection. The whole study was approved by the Ethical Committee of Psychological Research of Padua University protocol 3228.

**Statistical analysis.** *Statistical Package for Social Science* (SPSS) 25.0 software was used. To evaluate the equal distribution of the three main frequency levels (low frequency, medium frequency, and high frequency) we recoded each item into a 3-point scale and used the Chi-square test [24].

**Results.** Sixty-three items of the questionnaire fitted an equidistributional pattern and were selected to be submitted in the second phase of the study to be rated on a 5-point Likert scale ranging from 1 (never) to 5 (always). They assessed the degree of physical, emotional, and relationship pleasure and the satisfaction derived from orgasm. We named it the Orgasmic Perception Questionnaire (OPQ) to be found in the S1 File (63-item OPQ).

## Stage 2: Exploratory factorial analysis

### Methods

**Participants.** The sample for the EFA was composed of 699 participants of whom 25 omitted more than 10% of OPQ items and were eliminated from the analyses. A total of 674 Italian adults (34.1% men, 65.9% women) composed the final sample. Their socio-demographical information is presented in Table 1.

**Measures.** *Background questionnaire*. Information about age, gender, nationality, civil status, education level, employment, sexual orientation, relationship, and last sexual experience was asked.

*The Orgasmic Perception Questionnaire (OPQ)*. We used the 63-item OPQ (available in the S1 File) to be rated on a 5-point Likert scale ranging from 1 "never" to 5 "always", with the possibility to answer "I have never had a sexual partner" or "I don't know/ I don't remember".

**Procedure.** Participants answered the questionnaires and previously signed the informed consent. The collection of personal data provided for the possibility of not replying, by marking the option "I prefer not to reply". The questionnaires were administered to the general Italian population: 251 questionnaires on paper and 423 online, using the SurveyMonkey program. Participants were recruited online through personal contacts or calls on social media or in public places. The inclusion criteria included being older than 18 years and Italian mother tongue. Data were collected from May 2019 to January 2020 anonymously, so that authors had no access to information that could identify individual participants during or after data collection.

**Statistical analysis.** Exploratory Factor Analysis was performed using the principal axis extraction method and Oblimin rotation method.

**Results.** Five factors emerged explaining 44.01% of the total variance (see Table 2). According to the content of the items, the factors were named as follows: Factor 1 "Ecstasy" contained 16 items about the positive sensations experienced during orgasm; Factor 2 "Contractions" included 12 items about the perceptions and physical reactions of orgasm; Factor 3 "Wellbeing" contained 8 items about sensations characterized by mental and physical

**Table 2. Variance of the EFA.**

| Factor | N item | Variance % |
|---|---|---|
| 1. Ecstasy | 16 | 23.25% |
| 2. Contractions | 12 | 8.45% |
| 3. Wellbeing | 6 | 4.95% |
| 4. Power | 8 | 4.10% |
| 5. Sensations | 5 | 3.26% |
| **Total** | | **44.01%** |

**Table 3. Cronbach's alphas (N item).**

| Factor | EFA Sample | CFA Sample | Convergent Validity Sample | Divergent Validity Sample | Reliability Sample |
|---|---|---|---|---|---|
| Ecstasy | .89 (16) | .90 (16) | .90 (16) | .92 (16) | .90 (16) |
| Contractions | .91 (12) | .90 (12) | .92 (12) | .89 (12) | .89 (12) |
| Wellbeing | .71 (6) | .73 (6) | .75 (6) | .75 (6) | .76 (6) |
| Power | .79 (8) | .79 (8) | .83 (8) | .82 (8) | .74 (8) |
| Sensations | .70 (5) | .67 (5) | .66 (5) | .65 (5) | .61 (5) |

relaxation; Factor 4 "Power" included 8 items related to personal feelings such as the sense of power, and Factor 5 "Sensations" included 5 items of physical sensations such as the perception of heat on the face and the body.

In this phase, 10 items were removed because their commonalities were too low (less than .20) and the analysis was redone with 53 items. Next, 5 items were removed because all their factor loadings were too low (less than .30), and one item was removed because it was bi-factor with the higher factor loading in the factor with less coherence. Instead, we kept 3 bi-factor items because their higher factor loading was in the factor with more coherence. The final version of the test comprised 47 items.

Cronbach's alpha for the factors of Contraction was excellent, for the factor of Ecstasy was good, while for the factors of Wellbeing, Power, and Sensations these values were adequate (see Table 3). Factor loadings are reported in S1 Table of S1 File.

## Stage 3: CFA and other psychometric properties

### Method

**Participants.** For the CFA, the sample was composed of 1,100 participants (41% women, 59% men). Men aged from 18 to 63 (M = 26.40, SD = 7.89) while women were aged from 18 to 59 (M = 25.84, SD = 7.48), with no significant difference ($t$(1098) = 1.19, *n.s.*). The sample for the convergent validity was composed of 179 Italian adults with ages ranging between 18 and 51 years (*M* = 24.39, *SD* = 5.21). 66.5% identified themselves as female (*M* = 23.96, *SD* = 5.19) and 32.4% as male (*M* = 25.28; *SD* = 5.47), 2 people indicated "other." For the divergent validity, the sample was 198 Italian adults aged between 19 and 59 years (*M* = 26.20, *SD* = 6.74); 68.7% were female (*M* = 26.48; *SD* = 7.11) and 30.3% male (*M* = 25.60; *SD* = 5.96), while 2 people indicated "other". For reliability analysis, the sample consisted of 185, including 143 women (77.3%) and 42 men (22.7%), aged 19 to 51 years (*M* = 23.16; *SD* = 4.51). Data were collected from May 2020 to May 2022. The socio-demographic characteristics of the different samples are presented in Table 1.

**Measures.** *Background questionnaire*. Information about age, gender, nationality, civil status, education level, employment, sexual orientation, relationship, and last sexual experience was asked.

*The Orgasmic Perception Questionnaire (OPQ)*. We used the 47-item OPQ (available in the S1 File) to be rated on a 5-point Likert scale ranging from 1 "never" to 5 "always", with the possibility to answer "I have never had a sexual partner" or "I don't know/ I don't remember".

*Orgasm Rating Scale (ORS)*. The ORS [11] is a self-reported measure that evaluates the multidimensional subjective experience of orgasm, distributed along 4 dimensions (affective, sensory, intimacy, and rewards). The 25-item version [22] was used, with a 6-point Likert scale, where 0 indicates "does not describe it at all" and 5 "describes it perfectly". We translate it into Italian from both the Spanish and the English versions, comparing translations and using the back translation procedure. Cronbach's alpha ranged from .92 to .67.

*Balanced Inventory of Desirable Responding (BIDR).* The 16-item version of the BIDR 6 ([25]; Italian validation [26]) is a questionnaire assessing social desirability distributed along 2 dimensions (self-deceptive enhancement and impression management).

**Procedure.** Participation in the study consisted of completing a self-reported questionnaire that contained the mentioned instruments. The procedure was the same as for stage 2 but for using both Survey Monkey and Qualtrics programs to collect data. Only 80 questionnaires were administered on paper. We decide to use different samples to assess convergent and divergent validity and reliability to minimize administration time: too long administration times could have affected the tendency for less truthful responses to the last items presented. For the test-retest reliability, we use a longitudinal design, where all participants had to fill OPQ again a one-month distance from the first administration. Data were collected from May 2020 to January 2022 anonymously, so that authors had no access to information that could identify individual participants during or after data collection.

**Statistical analysis.** The S*tatistical Package for Social Science* (SPSS) software version 25.0 was used for all other analyses except for the CFA.

A multigroup Confirmatory Factor Analysis was conducted to test invariance between men and women, using lavaan package [27] in the R environment (version 4.2.2; R Core Team, 2022). The weighted least squares mean adjusted estimation method was used (WLSM), appropriate for ordinal scales [28]. To assess the adequacy of the model, we used the following fit index: Comparative Fit Index (CFI) and Tucker-Lewis index (TLI) values above .95, Standardized Root Means Square Residual (SRMR), and root mean square error of approximation (RMSEA) values lower than .08 [29]. Factorial invariance was gradually analyzed at six levels: configural invariance, metric invariance, scalar invariance, factor variance invariance, factor covariance invariance, and means invariance. To accept models' equivalence for the different levels, a change in the CFI that equals or exceeds .01 is considered to adopt the less limited model and to reject the most restrictive one [30].

Bivariate Pearson's correlations were tested between factors of the OPQ for the total sample and divided by gender. To assess gender differences in OPQ factors a MANCOVA was conducted with age as a covariate, using Bonferroni's correction. Reliability was tested by Cronbach's alpha: this coefficient varies between 0 and 1, with values close to 1 implying good homogeneity of the items (values > .70 indicate good internal reliability); values > 0.90 are considered excellent indicators, and values between 0.90 and 0.80 indicate good internal consistency, values between 0.80 and 0.70 indicate adequate internal consistency and values between 0.70 and 0.60 indicate sufficient internal consistency [31].

To verify test-retest reliability, convergent and divergent validity evidence were tested using the Student's t-test and the Pearson correlation. Values < .20 indicate a very small, correlation, values between .20 and .39 a small one, values between .40 and .59 an average one, values between .60 and .79 a strong one, and values > .80 a very strong one [32].

## Results

**CFA.** The fit indexes point out that the model of configural invariance was acceptable since RMSEA and SRMR were lower than .08 while CFI and TLI were close to .95 (see Table 4). All invariance steps are satisfied but the last one, dealing with factors means invariance (where Change CFI was higher than 0.01), since females had higher scores than males in all factors.

**Correlations.** Both the overall correlations and the correlations divided by gender between the factors were positive and significant, except for the correlation between the Contractions and Wellbeing Factors in women, which was very small and not significant (see Tables 5 and 6).

**Table 4. Measurement invariance across gender.**

| Model | $\chi^2$ | Df | $\chi^2$/df | RMSEA | GFI | CFI | TLI | SRMR | Change $\chi^2$ | Df | Change CFI |
|---|---|---|---|---|---|---|---|---|---|---|---|
| Configural Invariance | 12671.89 | 2840 | 4.46 | 0.079 | 0.945 | 0.940 | 0.937 | 0.077 | | | |
| Metric Invariance | 13419.14 | 2890 | 4.64 | 0.081 | 0.943 | 0.936 | 0.934 | 0.079 | 747.24 | 50 | 0.004 |
| Scalar Invariance | 13968.17 | 3050 | 4.58 | 0.081 | 0.940 | 0.933 | 0.935 | 0.077 | 549.03 | 160 | 0.003 |
| Factor Variance Invariance | 14005.60 | 3055 | 4.58 | 0.081 | 0.940 | 0.933 | 0.935 | 0.077 | 37.43 | 5 | 0.000 |
| Factor Covariance Invariance | 14109.98 | 3065 | 4.60 | 0.081 | 0.940 | 0.933 | 0.935 | 0.077 | 104.38 | 10 | 0.000 |
| Means Invariance | 17898.71 | 3070 | 5.83 | 0.094 | 0.923 | 0.910 | 0.913 | 0.077 | 3788.73 | 5 | 0.023 |

$\chi^2$ = chi square; df = degrees of freedom; RMSEA = root mean square error of approximation; GFI = Goodness of Fit Index; CFI = Comparative Fit Index; TLI = Tucker-Lewis Index; SRMR = standardized root mean square residual.

**Gender differences.** With the MANCOVA at the multivariate level, a significant effect of gender ($F(5,1093) = 68,26$; $p < .001$; $\eta^2$ partial = .24) and age ($F(5,1093) = 3,20$; $p = .007$; $\eta^2$ partial = .01) was found. At the univariate level, it was found that women scored significantly higher than men in three OPQ factors: $F(5,1093) = 25,11$; $p < .001$; $\eta^2$ partial = .02 for Ecstasy; $F(5,1093) = 219,54$; $p < .001$; $\eta^2$ partial = .17 for Contractions; $F(5,1093) = 214,70$; $p < .001$; $\eta^2$ partial = .17 for Sensations; and that older participants scored higher than younger ones on OPQ factor Ecstasy: $F(5,1093) = 10,84$; $p = .001$; $\eta^2$ partial = .01).

**Reliability.** Table 3 shows the internal consistency of the OPQ, which was tested by calculating Cronbach's alpha value for the total scale and each subscale. The total scale showed excellent internal consistency, Cronbach's alpha was .92 for the total sample as well as for men and women separately. Furthermore, the reliability was excellent for the Ecstasy and Contractions factors adequate for the Power and Wellness factors, and sufficient for the Sensations factor.

Test-retest reliability was good for all factors but the Ecstasy factor ($t(184) = 2.73$; $p = .007$; $d = .47$) wall other paired sample t-test values being not significant (see S2 Table in S1 File). All Pearson's correlations between the first and the second administration were strong or very strong and significant at the .01 level (.810 for F1 Ecstasy; .820 for F2 Contractions; .644 for F3 Wellbeing; .785 for F4 Power; and .756 for F5 Sensations).

**Convergent and divergent validity.** Many significant positive correlations between OPQ factors and ORS factors has been found (see Table 7). Small significant positive correlations emerged between the BIDR 6 SDE subscale and some OPQ factors, while no significant correlation emerged between the BIDR 6 IM subscale and the OPQ factors (see Table 7). Cronbach's alphas of these sample are reported in Table 3.

## Discussion

This study aimed to present the validation of a new Italian measure developed to assess sensations, emotions, and perceptions experienced during orgasm, the Orgasmic Perception

**Table 5. Bivariate correlations between factors of the OPQ for the total sample.**

| | 1 | 2 | 3 | 4 | 5 |
|---|---|---|---|---|---|
| 1. Ecstasy | 1 | | | | |
| 2. Contractions | 0.403** | 1 | | | |
| 3. Wellbeing | 0.422** | 0.131** | 1 | | |
| 4. Power | 0.575** | 0.284** | 0.439** | 1 | |
| 5. Sensations | 0.411** | 0.470** | 0.204** | 0.391** | 1 |

** correlation is significant at the 0.01 level (2-tailed)

**Table 6. Bivariate correlations between factors of the OPQ divided by gender.**

|  | 1 | 2 | 3 | 4 | 5 |
|---|---|---|---|---|---|
| 1. Ecstasy | 1 | .381** | .441** | .537** | .388** |
| 2. Contractions | .381** | 1 | .135** | .261** | .310** |
| 3. Wellbeing | .400** | .089** | 1 | .432** | .239** |
| 4. Power | .594** | .276** | .437** | 1 | .395** |
| 5. Sensations | .390** | .407** | .154** | .381** | |

The values below the diagonal belong to women and the values above the diagonal belong to men

** correlation is significant at the 0.01 level (2-tailed)

Questionnaire (OPQ), an instrument that evaluates different aspects of the subjective experience of orgasm. The necessity of this work stems from the lack of a standardized measure to specifically assess complete descriptions of orgasm experience since there are no previous instruments for examining complete descriptions of the bodily sensations, emotions, feelings, and perceptions experienced during orgasm for both men and women.

The multigroup confirmatory analysis, testing invariance between men and women, showed that the model that emerged from the exploratory analysis, consisting of the five factors of Ecstasy, Contractions, Wellbeing, Power, and Sensations, is acceptable. OPQ internal consistency was found to be satisfactory.

From the test of the effects between the subjects, a significant multivariate effect of both age and gender on the factors was observed. Univariate tests reveal the effect of age in the Ecstasy factor, with a too-small size effect to be considered. Univariate tests reveal the effect of gender in all OPQ factors, with a large effect size only for the Contractions and Sensations Factors, the others having a too-small effect size to be considered. Both the Contractions factor and the Sensations factor refer to perceptions and physical reactions related to orgasm: the first includes vibrations, tremors, spasms, contractions, and muscle tension; the latter includes the

**Table 7. Correlations between OPQ, ORS, and BIRD factors.**

|  |  | ORS_F1 | ORS_F2 | ORS_F3 | ORS_F4 | BIDR_SDE | BIDR_IM |
|---|---|---|---|---|---|---|---|
| OPQ_F1 | Female | .495** | .502** | .101 | .282** | .271** | -.062 |
|  | Male | .361** | .426** | .277* | .314* | .209 | .569 |
|  | Total | .450** | .488** | .146 | .270** | .248** | .002 |
| OPQ_F2 | Female | .349** | .582** | -.020 | .159 | .071 | -.156 |
|  | Male | .153 | .310* | .045 | .069 | -.045 | -.125 |
|  | Total | .296** | .542** | .074 | .500 | .027 | -.053 |
| OPQ_F3 | Female | .368** | .252** | .206* | .523** | .232** | .040 |
|  | Male | .275* | .278* | .116 | .517 | .215 | .245 |
|  | Total | .341** | .270** | .184* | .501** | .217** | .116 |
| OPQ_F4 | Female | .538** | .396** | .162 | .258** | .198* | -.005 |
|  | Male | .460** | .470** | .302* | .036 | .293* | .195 |
|  | Total | .520** | .445** | .227** | .179* | .219** | .076 |
| OPQ_F5 | Female | .370** | .441** | .015 | .183* | .110 | -.107 |
|  | Male | .073 | .368** | .218 | .124 | .084 | -.041 |
|  | Total | .287** | .480** | .126 | .132 | .077 | .010 |

** correlation is significant at the 0.01 level (2-tailed)

* correlation is significant at the 0.05 level (2-tailed).

perception of heat on the face and body, shortness of breath, localized physical pleasure, and feeling the urge to scream. The results relating to the differences in the means of the scores for the genders on these factors are in contrast with the stereotype that would see the male orgasm as more intense than the female one and with the very few data in the literature that report a longer orgasm duration in men than in women [1]. The data that emerged suggests that, although reaching orgasm in women is neither a constant nor fundamental phenomenon in the sexual act, when it occurs it is experienced more strongly than in men. Further studies are needed to better understand this phenomenon, using both quantitative and qualitative studies.

From test-retest reliability, no significant difference was found between the first and the second administration except for a negligible effect on the Ecstasy factor, while all correlations between the two administrations were strong or very strong. For convergent validity OPQ Ecstasy factor correlated positively with the ORS Affective and Rewards Dimensions; OPQ Contractions factor with ORS Sensory Dimension; OPQ Wellbeing factor with ORS Affective, Intimacy Dimension, and Rewards Dimensions; the OPQ Power factor with ORS Affective Dimension and Sensory Dimensions; and OPQ Sensations factor with ORS Sensory Dimension. The expected significant correlation between the OPQ Wellbeing factor and the ORS Intimacy dimension was not found in men and was significant but very small in women: this can be explained because the OPQ Wellbeing factor relates to feelings characterized by tranquility, calmness, mental and physical relaxation, comfort, a sense of protection from the partner, and a desire to smile, while ORS Intimacy dimension refers to an orgasmic experience in a sexual relationship characterized by intimacy. Again, the absence of a correlation between the OPQ Wellbeing factor and the ORS Intimacy dimension in males might indicate that men experience orgasm as a sense of comfort and protection from their partner regardless of whether they are in an intimate and/or stable relationship, while for women the intimacy of the relationship might be important but not necessary. Some studies had associated romantic relationship status with orgasm frequency and sexual satisfaction, showing that women who had a stable partner [33], who were in a committed (as opposed to casual) relationship [34, 35], who were highly likely to marry their partner [34] or who had experienced love in their relationship [36] tended to achieve orgasm more frequently. In addition, Pedersen and Blekesaune [37] reported that married or cohabiting couples tended to be more sexually satisfied: a stable relationship can positively influence sexual satisfaction in women. Furthermore, the OPQ Wellbeing factor correlated significantly with the ORS Affective and Rewards Dimensions: the feelings experienced during orgasm and the pleasurable or rewarding sensations resulting from the orgasmic experience correlated with the feelings characterized by mental and physical relaxation. According to the evolutionary perspective, the orgasmic experience represents a form of gratification for the attainment of basic survival needs, allowing for the maintenance of a stable pair bond [38, 39]: this perceived gratification could lead to feeling protected by one's partner, experiencing positive feelings related to the orgasmic experience.

Divergent validity was only partially verified, as a tendency emerged to unintentionally provide socially desirable responses by women in all OPQ factors but the Contractions and Sensations ones, where no significant correlations emerged, and by men in OPQ Strength Factor. Social expectations play a crucial role in the orgasmic experience, prompting mostly women to fake orgasms to alleviate increased feelings of anxiety, self-consciousness, and physiological abnormality [40]. Men probably unconsciously feel obliged to emphasize the sensations of strength, such as feelings of power and euphoria, linked to the orgasmic experience, as they are socially accepted for they are characteristic of a common sense of "masculinity". Overall, the significant correlations were small, and therefore the OPQ could still be considered a useful tool both in clinical settings and in the general population.

Physiological response to being perceived as "sexual" requires subjective awareness on the part of the person experiencing it [3]: a psychophysiological phenomenon that can be explained similarly to orgasm is pain, which includes the experience of a cut in the skin and not the cut itself [41]. So, ejaculation and pelvic contractions must be perceived in an erotically pleasurable way to be identified as orgasm, otherwise, such phenomena could be compared to any autonomic response such as, for example, a sneeze [1]. The OPQ factors seem to overlap with Mah and Binik's [1] model, in which, as with the experience of pain, each dimension includes different components of the orgasmic experience: the OPQ Contractions and Sensations factors can be included in the Mah and Binik's sensory dimension (that encompasses contractile sensations, muscle tension and release, and thermal sensations); the OPQ Wellbeing factor in the Mah and Binik's affective dimension (that encompasses euphoria, intimacy, and love); and the OPQ Power and Ecstasy factors in the Mah and Binik's evaluative dimension (that encompasses intensity, pleasure, satisfaction, and pain).

### Limitations and future studies

The research presented some limitations. The average age of the participant was very young which made it not possible to investigate possible age effects. The sample sizes were not so large and there were different samples for the confirmatory analysis, and to assess the reliability, the convergent validity, and the divergent one. Furthermore, other biases might be present in the study due to the cross-sectional data collected at one specific moment and to the self-report nature of the study (e.g., under- or overreporting). In the future will be useful to test older people, run multigroup confirmatory analyses for gender and age, and integrate the study with qualitative data from focus groups.

### Conclusion

The OPQ (to be found in the S1 File 47-item OPQ) could be a useful tool in both clinical settings and research studies to investigate orgasmic experience in its totality, taking into account the sensations, images, emotions, and perceptions experienced by the person from both a physiological and psychological point of view. Such an instrument is needed to make up for the lack of a tool that adequately assesses the subjective perception of orgasm, considering both the physical sensations and psychological experiences related to this sexual phenomenon. In addition, OPQ is a useful scale for assessing subjective experiences of orgasm both in the context of a relationship with a sexual partner and in the context of self-eroticism. It can help to better understand the orgasm experience, which is a multidimensional one and varies from person to person and from moment to moment even in the same person [42].

Even if the OPQ is not meant to be a clinical assessment tool, it may be used in clinical settings to explore the orgasmic perception of the patient, in a qualitative more than quantitative way. We do not propose a cut-off or minimum and maximum values, since subjective orgasm descriptions vary considerably both across individuals and on different occasions by the same individuals [43]. The clinician can explore with the patient the reason for a potential "bad" score and perform either a psychoeducational intervention or sex therapy depending on the cause (inadequate sexual stimulation, anxiety, sociocultural and generational expectations, etc.).

### Supporting information

**S1 File.**
(DOCX)

## Acknowledgments

Thanks are due to Irene Vallotto, Michela Lazzaroni, and Arianna Lanza for helping with data collection.

## Author Contributions

**Conceptualization:** Marta Panzeri.

**Formal analysis:** Lucia Ronconi.

**Investigation:** Denise Mauro.

**Methodology:** Marta Panzeri, Lucia Ronconi.

**Project administration:** Marta Panzeri.

**Resources:** Marta Panzeri.

**Visualization:** Denise Mauro.

**Writing – original draft:** Marta Panzeri, Denise Mauro.

**Writing – review & editing:** Marta Panzeri, Ana Isabel Arcos-Romero.

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
