## [Decision Letter · Decision Letter 0]

3 May 2023

PONE-D-23-07333Development of the Italian version of the Orgasmic Perception Questionnaire (OPQ)PLOS ONE

Dear Dr. Panzeri,

Thank you for submitting your manuscript to PLOS ONE. After careful consideration, we feel that it has merit but does not fully meet PLOS ONE’s publication criteria as it currently stands. Therefore, we invite you to submit a revised version of the manuscript that addresses the points raised during the review process.

We look forward to receiving your revised manuscript.

Kind regards,

Asia Mushtaq, Ph.D.

Academic Editor

PLOS ONE

**Journal Requirements:**

2. Make sure that all information related to ethics statement for this submission is included in the Methods section of the manuscript.

**Additional Editor Comments (if provided):**

Result section need careful revision, specifically reporting of in EFA-factor loadings of all scale items, scree plot etc.

Reviewers' comments:

Reviewer's Responses to Questions

**Comments to the Author**

1. Is the manuscript technically sound, and do the data support the conclusions?

Reviewer #1: Partly

Reviewer #2: Partly

2. Has the statistical analysis been performed appropriately and rigorously? 

Reviewer #1: Yes

Reviewer #2: Yes

3. Have the authors made all data underlying the findings in their manuscript fully available?

Reviewer #1: No

Reviewer #2: No

4. Is the manuscript presented in an intelligible fashion and written in standard English?

Reviewer #1: Yes

Reviewer #2: No

5. Review Comments to the Author

Reviewer #1: Dear editor thank you for your invitation to review manuscript entitled “Development of the Italian version of the Orgasmic Perception Questionnaire (OPQ)”

Comment 1: Authors need to be explaining application development of the Orgasmic Perception Questionnaire (OPQ) in health primary care or healthy setting and if a person had a bad score or had a problem in one of its aspect, what intervention can be taken for it.

Comment 2: Please provide more information regarding scoring Orgasmic Perception Questionnaire (OPQ) (minimum and maximum)

Comment 3: Please the term of 63-item in whole manuscript.

Reviewer #2: Review assignment for PONE-D-23-07333 

1. The introduction is too long. It is better to be shorter

2. Please explain more about step 1 of the study and how to remove a large number of questions due to the equidistribution pattern by mentioning the reference to the equidistribution pattern.

3. Instead of using the term study 1, 2, and 3, use the term stage 1, 2, and 3. Because all the steps are part of a single study, the 3 studies are not separate. For the same reason, the code of ethics should be written for the entire study once because it is a single code.

4. In each part of the study, the descriptive characteristics of the participants are boringly explained in the text. It is better to report the descriptive characteristics of the participants of each stage of the study in a table.

5. The details of accessing the samples in public places and online and inviting them to participate in the study should be explained.

6. How was the sample of 674 selected for exploratory factor analysis?

7. How were “I prefer not to reply” answers managed during data analysis? Were samples with this answer excluded from the analysis?

8. Using other factor extraction methods, such as principal axis factoring or unweighted least squares, is better than the PCA method. Because PCA does not attempt to explain the underlying population factor structure of the data and makes the often, unrealistic, assumption that each variable is measured without error.

9. The percentage of explained variance for the designed questionnaire is low (39.83%). What solution to improve this number is considered by the researchers? Is the communality of the items checked? Removing variables with communality less than 0.5 may improve the variance

10. It is not correct to perform t-test and correlation test to check test-retest reliability. Suitable tests should be replaced.

11. In Table 4, the sensations factor is not written

12. In the abstract, it is written that the score of women is higher than that of men for only 2 factors, while in line 397, it is written the score of women was higher than that of men for all the factors. Be checked

13. What is the reason for choosing the BIDR questionnaire for divergent validity?

14. Convergent validity and correlation results between OPQ and ORS questionnaire factors in the entire sample and by gender were reported in the text, which is very long and boring and can be converted into a table.

15. Face and content validity has not been quantitatively investigated. It is better to check and report CVI and CVR indicators.

16. The article is very detailed and long and the contents are sometimes repetitive. For example, Cronbach's alpha has been explained in several places. The article can be made shorter by converting a part of the results into a table, removing duplicate and redundant content, and rewriting the article.

6. PLOS authors have the option to publish the peer review history of their article (what does this mean?). If published, this will include your full peer review and any attached files.

Reviewer #1: No

Reviewer #2: No

---

## [Author Response · Author response to Decision Letter 0]

9 Jun 2023

Dear Editor and Reviewers:

Thank you very much for giving us the opportunity of improving our manuscript entitled “Development of the Italian version of the Orgasmic Perception Questionnaire (OPQ)”. We highly appreciate all your comments and recommendations, as well as the positive feedback received and the good evaluation of the paper. We would like to thank you for your consideration of this manuscript. We have highlighted the changes in yellow within the revised document. 

The result section needs careful revision, specifically reporting of in EFA-factor loadings of all scale items, scree plot etc.

Response: We inserted 3 new tables (Table 1, 3 , and 7) and unified info in the result section, revising it. We reported EFA-factor loadings in the “supporting information” file in Table 1S:

Reviewer #1: Dear editor thank you for your invitation to review manuscript entitled “Development of the Italian version of the Orgasmic Perception Questionnaire (OPQ)”

Response: We would like to thank this reviewer for the overall appreciation of our manuscript. We hope that we have addressed all the comments to their satisfaction.

Comment 1: Authors need to be explaining application development of the Orgasmic Perception Questionnaire (OPQ) in health primary care or healthy setting and if a person had a bad score or had a problem in one of its aspect, what intervention can be taken for it.

Comment 2: Please provide more information regarding scoring Orgasmic Perception Questionnaire (OPQ) (minimum and maximum)

Response: Thanks for raising this very interesting point. We added:

“Even if the OPQ is not meant to be a clinical assessment tool, it may be used in clinical settings to explore the orgasmic perception of the patient, in a qualitative more than quantitative way. We do not propose a cut-off or minimum and maximum values, since subjective orgasm descriptions vary considerably both across individuals and on different occasions by the same individuals (43). The clinician can explore with the patient the reason for a potential "bad" score and perform either a psychoeducational intervention or sex therapy depending on the cause (inadequate sexual stimulation, anxiety, sociocultural and generational expectations, etc.).”

Comment 3: Please the term of 63-item in whole manuscript.

Response: We changed the wording throughout the paper to "63-item”.

Reviewer #2: Review assignment for PONE-D-23-07333

Response: We would like to thank this reviewer for their overall appreciation of our manuscript. We hope that we have addressed all their comments to their satisfaction.

1. The introduction is too long. It is better to be shorter

Response: We agree with this comment, and we deleted both the parts concerning instruments irrelevant to this study and the part about neuroimaging

2. Please explain more about step 1 of the study and how to remove a large number of questions due to the equidistribution pattern by mentioning the reference to the equidistribution pattern.

Response: Thanks for raising this point. We clarified the reason of our methodological choice adding: “We choose to not use quantitative content validity measures since we wished that the selection would be performed by data analysis and not by experts: there are still too many biases on orgasm that according to us would not allow experts to judge objectively. Moreover, we wished to select items that could represent different ways to experience orgasm more than items that represent how most people experience it, and that could emerge only from laypeople's judgment.”

And we quoted:

DeVellis R. F. (2017). Scale development, theory and applications. SAGE Publications, Thousand Oaks, CA.

3. Instead of using the term study 1, 2, and 3, use the term stage 1, 2, and 3. Because all the steps are part of a single study, the 3 studies are not separate. For the same reason, the code of ethics should be written for the entire study once because it is a single code.

Response: Thanks for this suggestion: we substituted the term study with the term stage, unifying data on participants, Cronbach's alpha, and the ethical committee code.

4. In each part of the study, the descriptive characteristics of the participants are boringly explained in the text. It is better to report the descriptive characteristics of the participants of each stage of the study in a table.

Response: Thank you for this suggestion, we now did this in Table 1.

5. The details of accessing the samples in public places and online and inviting them to participate in the study should be explained.

Response: We added:

“(…) through personal contacts or calls on social media”

6. How was the sample of 674 selected for exploratory factor analysis?

Response: We specified that 25 participants were eliminated since they omitted more than 10% of the OPQ items.

7. How were “I prefer not to reply” answers managed during data analysis? Were samples with this answer excluded from the analysis?

Response: This was true only for socio-demographical questions, so we just calculate the percentages of socio-demographical data on the total sample. We added:

“socio-demographical data are not available for the whole sample (see Table 1).” 

We found no reason for excluding these participants. Important information such as age or gender was reported by each participant .

8. Using other factor extraction methods, such as principal axis factoring or unweighted least squares, is better than the PCA method. Because PCA does not attempt to explain the underlying population factor structure of the data and makes the often, unrealistic, assumption that each variable is measured without error.

Response: We run a principal axis EFA, that gave different results.

9. The percentage of explained variance for the designed questionnaire is low (39.83%). What solution to improve this number is considered by the researchers? Is the communality of the items checked? Removing variables with communality less than 0.5 may improve the variance

Response: Thank you very much for this suggestion. Now the EFA explains 44% of the variance. Considering communality > .50 would have left us with only 15 items. Since globally the 5 factors and the selected items made sense to us, we decided to consider items with communality > .20.

10. It is not correct to perform t-test and correlation test to check test-retest reliability. Suitable tests should be replaced.

Response: According to our knowledge, t-test and correlation are usually used to check test-retest reliability (see DeVellis R. F. (2017). Scale development, theory and applications. SAGE Publications, Thousand Oaks, CA.) 

11. In Table 4, the sensations factor is not written

Response: thank you for noticing it: we added it.

12. In the abstract, it is written that the score of women is higher than that of men for only 2 factors, while in line 397, it is written the score of women was higher than that of men for all the factors. Be checked

Response: We added: “with an adequate effect size”

13. What is the reason for choosing the BIDR questionnaire for divergent validity?

Response: The BIDR, due to its psychometric properties and characteristics, is the usual questionnaire used for assessing divergent validity since it measures different constructs from the ones we are studying and it is validated in Italian.

14. Convergent validity and correlation results between OPQ and ORS questionnaire factors in the entire sample and by gender were reported in the text, which is very long and boring and can be converted into a table.

Response: We did it, please see Table 7

15. Face and content validity has not been quantitatively investigated. It is better to check and report CVI and CVR indicators.

Response: We added:

“We choose to not use quantitative content validity measures since we wished that the selection would be performed by data analysis and not by experts: there are still too many biases on orgasm that according to us would not allow experts to judge objectively. Moreover, we wished to select items that could represent different ways to experience orgasm more than items that represent how most people experience it, and that could emerge only from laypeople's judgment.”

16. The article is very detailed and long and the contents are sometimes repetitive. For example, Cronbach's alpha has been explained in several places. The article can be made shorter by converting a part of the results into a table, removing duplicate and redundant content, and rewriting the article.

Response: Thank you for this very helpful suggestion. We did it.

Marta Panzeri and coauthors

---

## [Editor Report · Decision Letter 1]

6 Jul 2023

Development of the Italian version of the Orgasmic Perception Questionnaire (OPQ)

PONE-D-23-07333R1

Dear Dr. Marta Panzeri

We’re pleased to inform you that your manuscript has been judged scientifically suitable for publication and will be formally accepted for publication once it meets all outstanding technical requirements.

Kind regards,

Asia Mushtaq, Ph.D.

Academic Editor

PLOS ONE

---

## [Editor Report · Acceptance letter]

19 Jul 2023

PONE-D-23-07333R1 

Development of the Italian version of the Orgasmic Perception Questionnaire (OPQ) 

Dear Dr. Panzeri:

I'm pleased to inform you that your manuscript has been deemed suitable for publication in PLOS ONE. Congratulations! Your manuscript is now with our production department. 

Kind regards, 

on behalf of

Dr. Asia Mushtaq 

Academic Editor

PLOS ONE